# The Causal Relationship Between Dietary Factors and the Risk of Intracranial Aneurysms: A Mendelian Randomization Analysis

**DOI:** 10.3390/biomedicines13030533

**Published:** 2025-02-20

**Authors:** Longyuan Li, Jiaxuan Li, Mei Chang, Xin Wu, Ziqian Yin, Zhouqing Chen, Zhong Wang

**Affiliations:** Department of Neurosurgery & Brain and Nerve Research Laboratory, The First Affiliated Hospital of Soochow University, Suzhou 215006, China; llyuan66@163.com (L.L.); lijiaxuan1622@163.com (J.L.); changmeino@163.com (M.C.); wuxin112340@163.com (X.W.); sudayzq@163.com (Z.Y.)

**Keywords:** dietary, fruit, alcohol, coffee, intracranial aneurysm, Mendelian randomization analysis

## Abstract

**Background:** Some studies have shown that dietary factors can influence the occurrence of intracranial aneurysms (IAs). This study aimed to investigate whether dietary factors and habits are associated with intracranial aneurysms using Mendelian randomization (MR) analysis. **Methods:** A two-sample MR study was conducted to evaluate the association of dietary factors with IAs. Heterogeneity was evaluated using Cochran’s Q test, and horizontal pleiotropy was assessed through MR-Egger regression and MR-pleiotropy residual sum and outlier (MR-PRESSO). **Results:** Fresh fruit intake (OR: 0.28, 95% CI: [0.13, 0.59]) was related to a decreased risk of IAs. Lamb/mutton intake may be associated with IAs, although the meta-analysis results were not significant (OR: 1.43, 95% CI: [0.27, 7.67]). Furthermore, MR analyses based on two aneurysm databases showed that alcoholic intake was not associated with IAs (alcoholic drinks per week: OR: 1.057, 95% CI: [0.788, 1.42]; OR: 0.509, 95% CI: [0.1665, 1.56]; alcohol intake frequency: OR: 1.084, 95% CI: [0.909, 1.29]; OR: 1.307, 95% CI: [0.814, 2.1]). Our results showed no causal relationship between coffee intake and IAs (OR: 1.149, 95% CI: [0.575, 2.3]; OR: 0863, 95% CI: [0.2979, 2.5]). Other dietary intakes were also found to have no causal relationship with IAs. **Conclusions:** This study found that fresh fruit intake was associated with a reduced risk of IAs. Lamb/mutton intake may be associated with IAs. However, other dietary factors, including alcohol intake and coffee intake, were found not to be associated with IAs.

## 1. Introduction

Subarachnoid hemorrhage (SAH) is an important subtype of stroke that predominantly occurs when a cerebral aneurysm ruptures. SAH caused by aneurysm rupture mainly affects people under 65 years old and has high mortality and morbidity rates [1,2]. The incidence of intracranial aneurysms (IAs) in the adult population is approximately 3.2% [3]. SAH resulting from IA rupture is associated with a 35% mortality rate within 3 months, and 50% of survivors suffer from severe neurological deficits [4]. These often include cognitive impairment, motor dysfunction, and a notable reduction in quality of life, collectively placing a substantial economic and social burden on healthcare systems, patients, and their families worldwide. IA rupture often occurs suddenly, and most patients are unaware of the presence of an aneurysm, so the high mortality rate caused by aneurysms is partly attributable to the lack of effective prevention methods [5,6]. Previous studies aimed to identify risk factors for IA formation, growth, and rupture, but these studies were limited to unchangeable risk factors such as gender, age, and family history [7]. This underscores the pressing need to explore modifiable lifestyle factors—especially dietary habits—that could potentially mitigate the risk of aneurysm development and rupture.

An increasing number of researchers have identified various dietary factors that may be associated with the development of aneurysms. One previous study suggested that consuming antioxidant vitamins could significantly lower the risk of IA formation, whereas the intake of alcohol and caffeine might increase it [8]. However, as this study was merely a review, its level of evidence was limited. Moreover, dietary factors may also influence the likelihood of IA rupture. Several clinical studies have indicated that coffee consumption could be an independent risk factor for increasing the risk of IA rupture [9,10]. In addition, a case–control study showed that consuming fat and meat more than four times per week may be associated with SAH, whereas drinking skimmed or reduced-fat milk and eating fruit more than four times per week has a protective effect against SAH [11]. These observations highlight the importance of further investigation into dietary patterns and specific nutrients that may either exacerbate or protect against aneurysm formation and rupture.

Therefore, understanding the impact of dietary factors on IAs is of great significance to clinicians and patients. Mendelian randomization (MR) is a statistical technique for identifying causal relationships. MR uses instrumental variables (IVs), specifically single-nucleotide polymorphisms (SNPs) that are strongly associated with the exposure, to investigate causality between the exposure and the outcome, avoiding the effects of confounders and reverse causality [12]. By leveraging genetic variants as proxies for dietary factors, MR can help determine whether specific dietary exposures truly contribute to aneurysm risk, rather than merely being correlated with it. However, to date, no study has employed an MR approach to elucidate the causal link between dietary factors and the development of aneurysms. To bridge this knowledge gap, we designed an MR analysis that incorporates data from multiple databases of patients with unruptured aneurysms, providing a robust platform to explore which dietary components may influence IA occurrence. Through this approach, we aim to offer clearer insights into the preventative potential of dietary modifications and ultimately contribute to reducing the global burden of SAH.

## 2. Methods

### 2.1. Data Source

In this study, we utilized a large public genome-wide association study (GWAS) dataset and MR analysis to detect the relationship between diet-related exposures, including dried fruit intake, salad/raw vegetable intake, cooked vegetable intake, cheese intake, processed meat intake, poultry intake, beef intake, non-oily fish intake, oily fish intake, pork intake, lamb intake, bread intake, cereal intake, alcoholic drinks per week, alcohol intake frequency, tea intake, coffee intake, fresh fruit intake, and IAs. The GWAS summary data of dietary factors were extracted from the UK Biobank by the IEU open GWAS project (https://gwas.mrcieu.ac.uk/; api: gwas-api.mrcieu.ac.uk/; accessed on 27 August 2024). The GWAS summary data of aneurysms were extracted from the European Bioinformatics Institute (EBI) and FinnGen Biobank by the IEU open GWAS project. The design of this MR analysis was based on the following assumptions: (1) the genetic variants used are independent of any confounding factors; (2) the genetic variants are directly and strongly associated with dietary factors; (3) the genetic variants used affect the development of IAs only by influencing dietary factors.

### 2.2. The Selection of IVs

IVs are used as a mediator between exposure factors and outcomes to explore the causal relationship between exposure and outcomes in MR analysis. Single-nucleotide polymorphisms (SNPs) were significantly associated with the exposure factors as instrumental variables. We used a strict threshold (kb = 10,000, r^2^ < 0.001) clustering procedure to ensure the independence of the selected SNPs, and SNPs with a significant correlation with the results (*p* < 5 × 10^−8^) were also discarded. The F statistic was used to ensure a strong correlation between the IVs and dietary factors. It is generally believed that an F statistic greater than 10 meets the requirements for a strong correlation [13].

### 2.3. The Statistical Analysis

The inverse variance weighting (IVW) model is the method with the strongest causality detection ability in two-sample MR analysis [14]. Thus, we used the IVW method to calculate major causal effects. Meanwhile, Cochran’s Q test was used to assess the heterogeneity of the IVW model, and Cochran’s Q test *p* < 0.05 indicated the presence of heterogeneity. In addition to the IVW model, we also applied MR-Egger, weighted median, simple mode, and weight mode for additional statistical analysis. MR-Egger can detect and correct potential horizontal pleiotropy, but the results may be affected by outlier genetic variables. Therefore, in the presence of horizontal pleiotropy, the MR-Egger slope is relatively effective as an estimate of MR [15]. The weighted median method provides better detection of causality under certain conditions because the weighted median method eliminates 50% of invalid IVs when assessing causality [14,16]. We performed pleiotropy tests and MR-PRESSO outlier tests. Once outliers were found, the corresponding SNPs were deleted. After removing SNPs, MR analysis was performed again. We also used the FDR method for correction; a corrected *p*-value less than 0.05 indicates statistical significance. A leave-one-out analysis was performed to assess whether removing individual SNPs had a significant impact on the results. The scatter plots showed the relationship between SNPs and exposure and outcomes, whereas the forest plots showed how individual instrumental variables affected the overall estimation of causality. All analyses were performed in R software (version 4.3.1) using the Two Sample MR and MR-PRESSO packages.

## 3. Results

Eighteen dietary exposure factors were used to analyze their causal relationship with IAs. The number of SNPs used in this analysis ranged from 8 to 99. The F statistic values were more than 10 (minimum = 32.5384, maximum = 76.2197). The number of people exposed ranged from 335,394 to 462,346. The GWAS database of results included European populations from the FinnGen and European biobanks, and there was little overlap between the populations involved in exposure and outcomes. More information on exposure and outcomes is shown in Table 1. This suggests that the IV used in our study meets the requirement for a strong association with exposure.

### 3.1. The Results of the MR Analysis

We conducted two MR analyses based on two IA GWAS databases. The results of the MR analysis using the EBI database showed a causal relationship between fresh fruit intake, lamb/mutton intake, and IAs in the IVW test. Fresh fruit intake was associated with a reduced risk of IAs (OR: 0.209, 95% CI: [0.0893, 0.491], *p* = 0.000327), while lamb/mutton intake may increase the risk of IAs (OR: 2.831, 95% CI: [1.0304, 7.78], *p* = 0.0436). Fresh fruit intake also resulted in the same conclusion in the weighted median model (OR: 0.123, 95% CI: [0.0325, 0.467], *p* = 0.002069). However, the MR-Egger model did not produce significant results (*p* > 0.05). In addition, alcoholic drinks per week, alcohol intake frequency, and coffee intake did not show a causal relationship with IAs for any method (using the IVW method: alcoholic drinks per week: OR: 1.057, 95% CI: [0.788, 1.42]; alcohol intake frequency: OR: 1.084, 95% CI: [0.909, 1.29]; coffee intake: OR: 1.149, 95% CI: [0.575, 2.3]. More details are shown in Figure 1 and Appendix A.

In the analysis using GWAS data of IAs from the FinnGen biobank, fresh fruit intake (OR: 0.721, 95% CI: [0.15803, 3.2]), lamb/mutton intake (OR: 0.4887, 95% CI: [0.0715, 3.34]), alcoholic drinks per week (OR: 0.509, 95% CI: [0.1665, 1.56]), alcohol intake frequency (OR: 1.307, 95% CI: [0.814, 2.1]), coffee intake (OR: 0.863, 95% CI: [0.2979, 2.5]), and other dietary factors were not causally associated with IAs in the IVW or other models. The details of the results are shown in Figure 2 and Appendix A. After FDR correction, fresh fruit intake was still associated with IAs when using the EBI database as an outcome. The results of FDR correction of the *p*-value are shown in Appendix A.

In addition, we conducted an MR-PRESSO test, and no outliers were detected. The heterogeneity test results revealed that no exposure factors showed heterogeneity (Cochrane’s Q test, *p* > 0.05). The details are provided in Appendix A. The leave-one-out plots, scatter plots, forest plots, and funnel plots are shown in Appendix A.

### 3.2. Meta-Analysis of the Two MR Analyses

We extracted the data from two different GWAS databases, and the MR analysis results of each database were not the same. Therefore, we also conducted a meta-analysis to draw more robust conclusions. Our meta-analysis results showed that fresh fruit intake is associated with a reduced risk of IAs (OR: 0.28, 95% CI: [0.13, 0.59]), and lamb/mutton intake may not have a causal relationship with IAs (OR: 1.43, 95% CI: [0.27, 7.67]). The details of the meta-analysis are shown in Figure 3.

### 3.3. The Results of MR Analysis in Asian Population

In this study, we aimed to investigate the relationship between different dietary factors and the risk of aneurysms, taking into account the potential influence of ethnic differences. We specifically conducted a Mendelian randomization (MR) analysis in an Asian population because individuals in this region may have distinct genetic backgrounds, dietary habits, and lifestyles (Figure 4). Moreover, most previous research has focused on Western populations, leaving limited evidence regarding Asian populations. Consistent with our earlier findings, this analysis suggests that consuming fresh fruit might provide a significant protective effect against intracranial aneurysms (IAs) (OR = 0.20, 95% CI = [0.066, 0.61]). This result indicates that the vitamins, minerals, and dietary fiber in fresh fruit could play a crucial role in preventing arterial wall pathology and mitigating inflammatory responses.

Conversely, we also observed that higher intake of processed meat and mutton may substantially increase the risk of IAs (processed meat OR = 3.29, 95% CI = [1.0552, 10.3]; mutton OR = 5.3, 95% CI = [1.46, 19.2]). This may be related to the elevated levels of saturated fatty acids, salt, or potential pro-inflammatory substances found in processed meats and red meat. Prolonged excessive consumption of these foods could impair vascular endothelial function and exacerbate inflammation, thereby promoting aneurysm formation and progression [17,18]. Notably, these associations appear to be more pronounced in Asian populations, suggesting that certain similar dietary risk factors may remain influential even in regions with different cultural dietary practices. The details of the MR analysis in asian population are shown in Figure 4.

## 4. Discussion

We pooled the results from two IA GWAS databases and found that fresh fruit intake was associated with an increase in the risk of IAs. Although our pooled analysis showed that lamb/mutton meat intake was not associated with IAs, a relationship between lamb/mutton intake and IAs cannot be completely excluded. In particular, cultural differences in cooking methods (e.g., high-temperature grilling versus stewing) or the use of certain seasonings may modify the impact of lamb/mutton on IA risk. Additionally, alcoholic drinks per week, alcohol intake frequency, poultry intake, non-oily fish intake, beef intake, pork intake, oily fish intake, processed meat intake, bread intake, cooked vegetable intake, cheese intake, tea intake, salad/raw vegetable intake, and coffee intake were not found to be associated with IAs in the MR analyses of the two databases, as well as in the pooled results.

The conclusions of this study are helpful in reducing the risk of IAs in high-risk groups by adjusting their dietary habits. In addition, clinicians can refer to our research conclusions to strengthen the health management of IA patients and encourage them to change their dietary habits (such as reducing mutton intake and increasing fresh fruit intake). Therefore, this study is of great significance for understanding the risk factors and protective factors of IAs. From a practical standpoint, implementing dietary guidance in routine clinical visits—especially for individuals with a family history of IAs or other vascular risk factors—could offer an additional layer of prevention.

Risk factors for IAs often include age, high blood pressure, smoking, and alcohol consumption [19]. A large number of previous studies have shown that alcohol consumption is related to the occurrence, growth, and rupture of IAs [8,20,21]. A previous study showed that the risk of aneurysms is positively related to the amount of alcohol consumed. In addition, the risk of aneurysm rupture was only related to current alcohol consumption, and past drinking habits did not increase the risk of aneurysms [8,21,22,23]. However, the study results of Cras et al. on unruptured IAs showed that although hypertension and smoking were still associated with IAs, alcohol consumption, physical activity, and diet quality were not associated with the presence of unruptured IAs [24]. This is partially consistent with the conclusion of our MR analysis, i.e., neither weekly drinking nor drinking frequency was significantly associated with the occurrence of aneurysms. In other words, while alcohol remains a well-documented vascular risk factor in many contexts, its direct link to IA formation might be more nuanced than previously believed, potentially involving dose thresholds, individual metabolism differences, or other lifestyle factors.

Unlike previous studies, MR analysis uses genetic variants (SNPs) as IVs, and the study results are not affected by confounding factors. Previous Mendelian randomization studies have examined the relationship between aneurysms and dietary factors, suggesting that alcohol may be associated with the rupture of aneurysms leading to subarachnoid hemorrhage (SAH) [25]. It is possible that alcohol’s role in aneurysm pathology is more directly tied to rupture mechanisms, such as acute blood pressure elevation or local inflammatory changes in the vessel wall, rather than initial aneurysm formation. Our study focuses more on the association between dietary factors and the probability of unruptured aneurysm formation, excluding patients with ruptured aneurysms. Considering that the selection of different ending databases may affect the result, we incorporated two aneurysm databases and conducted META analysis of the two endings, which made our results more reliable. In addition, we still used the aneurysm data of the Asian population to verify our results, proving that this conclusion still exists in different races. We believe that the possible explanations for the differences between the conclusions of this MR study and those of previous studies on the relationship between alcohol and IAs are as follows: First, observational studies may still be affected by other confounding factors, although the confounding factors may have been adjusted. Second, there may be differences in the patients included in different studies. The conclusion of a previous retrospective study that only included unruptured aneurysms is consistent with our MR study, which showed that alcohol consumption is not related to the presence of IAs [24]. Other studies have shown that the amount and intensity of alcohol consumption are significantly related to IA rupture [23]. Therefore, the inclusion of ruptured aneurysms may have had an impact on the results. More prospective and MR studies are needed in the future to further reveal the relationship between alcohol intake and IAs. Although, according to our conclusion, there is no causal relationship between alcohol consumption and IAs, alcohol consumption remains a risk factor in patients with IAs. This indicates that clinical recommendations for reducing or abstaining from alcohol are still relevant for patients already diagnosed with IAs, to minimize the possibility of exacerbating underlying vascular weaknesses or triggering rupture.

The MR analysis results showed that fresh fruit can reduce the risk of intracranial aneurysms. Fresh fruits are rich in antioxidant vitamins with anti-inflammatory properties, reducing the production of pro-inflammatory eicosanoids and thus preventing the formation and rupture of IAs [8,26]. Additionally, the high fiber content in many fruits may help regulate blood pressure and overall vascular health, further contributing to lower IA risk [27]. Although the results from two of the MR analyses showed that mutton consumption was associated with a higher risk of IAs, the meta results of two European populations MR analyses showed that there was no causal relationship between mutton and IAs. Previous studies have shown that meat is associated with a higher risk of IAs; however, no studies have examined the association between lamb and IAs, so this conclusion needs to be interpreted with caution. In addition, we found that processed meat showed different results on the occurrence of IAs in different populations. This may be due to differences in the raw materials and additives of processed meat in different regions. Process-related methods (e.g., curing, smoking) could influence the levels of saturated fats, sodium, or nitrates, each of which may affect vascular integrity and inflammation. As shown in our study, different meats may lead to different aneurysm risks. Future research needs to further focus on the effects of different additives or trace elements on IAs.

Our study has several limitations. First, despite using GWAS data from the EBI and the Finnish databases, the data sample sizes for IAs are still small, and studies with larger populations are needed to replicate the results obtained. Additionally, although we selected exposure factors and outcome factors from different countries, effectively avoiding the problem of overlapping sample sizes, the analysis was mainly based on European and Asian populations, and there is still a need to pay attention to some possible differences between different populations in the future. In addition, as we were unable to obtain further individual-level data from the GWAS database, we used summary statistics rather than individual-level data, which prevented us from performing subgroup analyses for potential confounders such as gender, age, and others. Previous studies have shown that gender and age affect the occurrence of aneurysms, and the female gender is a risk factor for IAs [24]. Furthermore, we were unable to further segment different types of dietary intake or distinguish the effects of different dietary combinations.

## 5. Conclusions

This study found that fresh fruit intake was associated with a reduced risk of intracranial aneurysms. Lamb/mutton and processed meat intake increased the risk of intracranial aneurysms in Asian populations. However, other dietary factors, including alcohol intake and coffee intake, were found not to be associated with IAs. More MR analyses or prospective clinical trials are needed to explore the relationship between dietary habits and IAs.

## Figures and Tables

**Figure 1 biomedicines-13-00533-f001:**
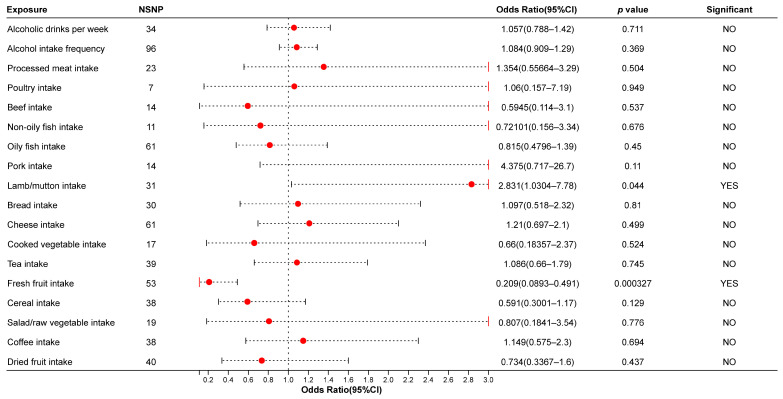
The results of the MR analysis using the IVW method (ebi-a-GCST90018815). NSNP: Number of Single Nucleotide Polymorphisms (SNPs); Red dots in figure: point estimate.

**Figure 2 biomedicines-13-00533-f002:**
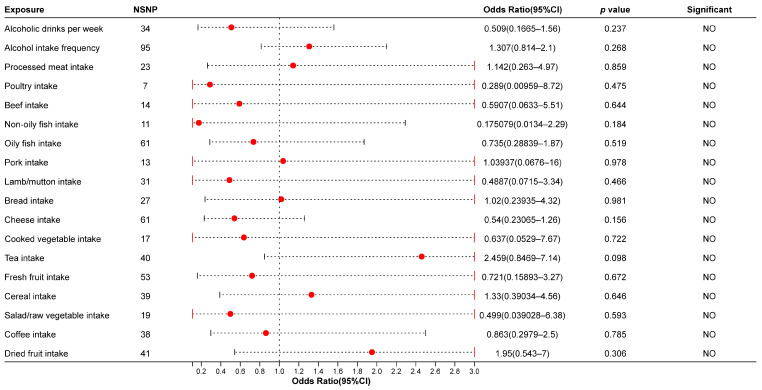
The results of the MR analysis using the IVW method (finn-b-I9_ANEURYSM). NSNP: Number of Single Nucleotide Polymorphisms (SNPs); Red dots in figure: point estimate.

**Figure 3 biomedicines-13-00533-f003:**
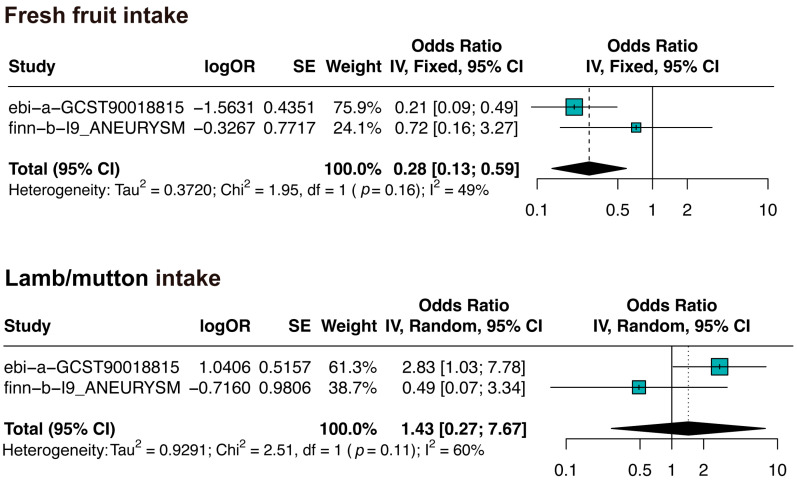
Meta-analysis of two MR analyses. Squares represent the effect estimates from individual studies, with the size of each square being proportional to the study’s weight in the meta-analysis. The horizontal lines indicate the 95% confidence intervals, and the diamond represents the pooled effect estimate along with its 95% confidence interval. df: degrees of freedom.

**Figure 4 biomedicines-13-00533-f004:**
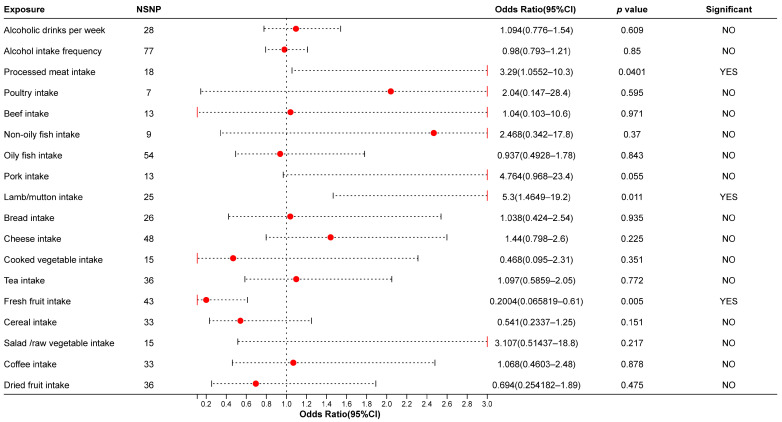
Detail of MR analysis in Asian Population. NSNP: Number of Single Nucleotide Polymorphisms (SNPs); Red dots in figure: point estimate.

**Table 1 biomedicines-13-00533-t001:** Information of the exposure and outcome datasets.

GWAS ID	Exposure or Outcome	Identified SNPs	Participants Included in the Analysis	F Statistics	R^2^
ieu-b-73	Alcoholic drinks per week	35	335,394	76.2197	0.0001458
ukb-b-5779	Alcohol intake frequency	99	462,346	52.7013	0.0001139
ukb-b-6324	Processed meat intake	23	461,981	38.5360	8.34 × 10^−5^
ukb-b-8006	Poultry intake	8	461,900	32.5384	7.044 × 10^−5^
ukb-b-2862	Beef intake	17	461,053	41.4726	8.99 × 10^−5^
ukb-b-17627	Non-oily fish intake	11	460,880	44.8021	9.72 × 10^−5^
ukb-b-2209	Oily fish intake	63	460,443	44.9179	9.75 × 10^−5^
ukb-b-5640	Pork intake	14	460,162	37.6855	8.19 × 10^−5^
ukb-b-14179	Lamb/mutton intake	32	460,006	39.5946	8.61 × 10^−5^
ukb-b-11348	Bread intake	32	452,236	41.8138	9.25 × 10^−5^
ukb-b-1489	Cheese intake	65	451,486	39.0172	8.64 × 10^−5^
ukb-b-8089	Cooked vegetable intake	17	448,651	37.5840	8.38 × 10^−5^
ukb-b-6066	Tea intake	41	447,485	68.8119	0.0001358
ukb-b-3881	Fresh fruit intake	55	446,462	45.9501	0.0001029
ukb-b-15926	Cereal intake	43	441,640	45.0296	0.0001019
ukb-b-1996	Salad/raw vegetable intake	22	435,435	38.3330	8.80 × 10^−5^
ukb-b-5237	Coffee intake	40	428,860	72.7115	0.0001694
ukb-b-16576	Dried fruit intake	43	421,764	41.8987	9.93 × 10^−5^
ebi-a-GCST90018815	Cerebral aneurysm	NA	473,683	NA	NA
finn-b-I9_ANEURYSM	Cerebral aneurysm, nonruptured	NA	NA	NA	NA

SNPs: single-nucleotide polymorphisms; GWAS: genome-wide association study; N/A: not applicable.

## Data Availability

All data generated or analyzed during this study are included in this published article and its Appendix A.

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
