# Peer review of "The Causal Relationship Between Dietary Factors and the Risk of Intracranial Aneurysms: A Mendelian Randomization Analysis"

_biomedicines, 2025, doi:10.3390/biomedicines13030533_

Round 1
Reviewer 1 Report
Comments and Suggestions for Authors
Could the authors explain the new of this MS. from the following published one:
https://doi.org/10.1080/1028415X.2024.2403894
Comments on the Quality of English LanguageThe English could be improved to more clearly express the research.
Author Response
Comments1: Could the authors explain the new of this MS. from the following published one:
https://doi.org/10.1080/1028415X.2024.2403894
Response 1: Thank you for your comment. We believe our study offers several innovations compared to the previous work. First, our conclusions differ from those of the earlier study, which we attribute to the different endpoints selected in our research. Specifically, our study focuses solely on non-ruptured aneurysms and primarily investigates the risk factors for aneurysm formation, rather than rupture. We believe there may be distinct differences between the risk factors for aneurysm formation and those for rupture. For instance, our findings do not identify alcohol consumption as a risk factor for aneurysm formation, which is interesting because alcohol is frequently cited as a risk factor for aneurysm rupture.
Additionally, recognizing that the choice of endpoint data can influence the results, we incorporated two aneurysm databases and conducted a meta-analysis to combine the data from both endpoints. This approach enhances the robustness of our findings. Furthermore, we validated our results using data from the Asian population, demonstrating that our conclusions hold across different racial groups.
Finally, we focused on the results that were significant after applying the FDR (false discovery rate) correction, which strengthens the rigor of our conclusions.
We have addressed the differences between our study and the previous research in the discussion section (lines 195-201).
Comments2: The English could be improved to more clearly express the research.
Response1: Thank you for your kind reminder. We have now completed the English language editing of the entire manuscript.
Reviewer 2 Report
Comments and Suggestions for Authors
The study conducted by Xin Wu et al., entitled “Causal Relationship Between Dietary Factors and Risk of Intracranial Aneurysms: A Mendelian Randomization Analysis” appears to be an interesting as highlighted the relationship between dietary factors and the incidence of intracranial aneurysms. The authors findings suggest that fresh fruit intake is associated with a reduced risk of intracranial aneurysms (IAs), whereas lamb/mutton intake may be associated with an increased risk of IAs. Additionally, the study found that other dietary factors, such as alcohol and coffee intake, were not associated with IAs.
The manuscript is well-designed, and the methodology employed is suitable for the research question. However, similar studies have been conducted previously, including one by Linazi G et al. in 2024 (Linazi, G., Maimaiti, A., Abulaiti, Z., Adili, N., Guan, J. and Abulaiti, A., 2024. Dietary factors and the incidence of intracranial aneurysms: a Mendelian randomization research. Nutritional Neuroscience, pp.1-9).
The authors do not cite these earlier studies, nor do they compare their analysis with previously reported data on the same topic. As a result, the manuscript lacks sufficient novelty.
I would like to suggest that the authors cite relevant articles in this area and discuss the differences and similarities between their findings and those of previous studies. Furthermore, the paper requires careful proofreading to eliminate the numerous typographical errors present. The authors mention that some data are included in a supplementary file, but this file is missing.
In brief, I recommend the authors address these issues by:
1. Conducting a thorough literature review to identify and cite relevant previous studies.
2. Comparing their findings with those of earlier studies to highlight the novelty and significance of their research.
3. Ensuring that the manuscript is carefully proofread to eliminate typographical errors.
4. Providing the missing supplementary file or incorporating the relevant data into the main manuscript.
Author Response
Comments 1: Conducting a thorough literature review to identify and cite relevant previous studies.
Response 1: Thank you for your suggestion. We have conducted a comprehensive literature review and have included relevant citations throughout the manuscript where appropriate. Additionally, we compared our findings with those of previous studies in the discussion section (lines 195-201).
Comments 2: Comparing their findings with those of earlier studies to highlight the novelty and significance of their research.
Response 2: We appreciate your comment. Our study presents several innovations. First, our conclusions differ from those of the previous study, which we believe is due to the different endpoints we selected. Specifically, our research focuses exclusively on non-ruptured aneurysm data and examines the risk factors for aneurysm formation, rather than rupture. We hypothesize that the risk factors for aneurysm formation may differ from those for rupture. For example, our study did not find alcohol consumption to be a risk factor for aneurysm formation, which is intriguing since alcohol is commonly considered a risk factor for aneurysm rupture.
Furthermore, to account for the potential impact of endpoint selection on the results, we incorporated data from two aneurysm databases and conducted a meta-analysis. This approach strengthens the reliability of our findings. Additionally, we validated our results using aneurysm data from the Asian population, which supports the generalizability of our conclusions across different racial groups.
Finally, we selected only the results that were significant after applying the FDR (false discovery rate) correction, ensuring that our conclusions are more robust and rigorous.
Comments 3: Ensuring that the manuscript is carefully proofread to eliminate typographical errors.
Response 3: Thank you for your reminder. We have thoroughly proofread the manuscript to ensure that it is free from typographical errors.
Comments 4: Providing the missing supplementary file or incorporating the relevant data into the main manuscript.
Response 4: Thank you for pointing this out. We have addressed this issue and have either provided the missing supplementary file or incorporated the relevant data into the main manuscript as necessary.
Reviewer 3 Report
Comments and Suggestions for Authors
The presented manuscript by Xin Wu et al. is an attempt to find a relationship between the risk of intracranial aneurysm (a serious disease with high mortality and disability) and some modifiable factors, such as diet. This topic is certainly important, but the approach chosen by the authors seems quite controversial. Indeed, the question arises as to how applicable to real life is the consideration of the effect of diet on intracranial aneurysms without taking into account such non-genetic factors as age, physical activity, tobacco use, as well as hormonal drugs and other medications. Is it possible to compare the effect of genetic factors or lifestyle with the effect of diet? In addition, the methodology for selecting data for analysis and cleaning the primary data from concomitant factors is not described in sufficient detail, which makes it difficult to assess the correctness of this approach.
Author Response
Comments 1: The presented manuscript by Xin Wu et al. is an attempt to find a relationship between the risk of intracranial aneurysm (a serious disease with high mortality and disability) and some modifiable factors, such as diet. This topic is certainly important, but the approach chosen by the authors seems quite controversial. Indeed, the question arises as to how applicable to real life is the consideration of the effect of diet on intracranial aneurysms without taking into account such non-genetic factors as age, physical activity, tobacco use, as well as hormonal drugs and other medications. Is it possible to compare the effect of genetic factors or lifestyle with the effect of diet?
Response 1: Thank you for your thoughtful comment. We agree that the relationship between diet and intracranial aneurysms is complex, and the consideration of modifiable factors, such as age, physical activity, tobacco use, and hormonal medications, is crucial. However, in our study, we focused on exploring the potential causal relationship between diet and aneurysm risk through Mendelian Randomization (MR). MR utilizes instrumental variables, particularly single nucleotide polymorphisms (SNPs), that are strongly associated with exposure (in this case, diet) to help establish causal links, thus minimizing the influence of confounding factors.
We acknowledge that aneurysm development is influenced by a variety of factors, including genetics and lifestyle. While we have focused on genetic aspects in this study, we recognize the importance of considering other modifiable factors, such as those you mentioned. This is indeed a topic for future research, and we aim to investigate the broader context, incorporating additional lifestyle and environmental factors, to fully understand their roles in aneurysm risk.
Regarding the methodology for selecting and cleaning the primary data, we appreciate your feedback. The exposure and outcome data used in our study were selected from the same population, but we made an effort to source data from different regions to increase the generalizability of our findings. Moving forward, we will also prioritize the use of newer databases with larger populations to reduce potential biases and enhance the robustness of our analysis.
Reviewer 4 Report
Comments and Suggestions for Authors
Dear Authors,
1. The sample size of the GWAS databases used in the study is relatively small. Therefore, further research should use larger datasets to enhance the reliability of the results.
2. The study primarily focuses on data from European populations. Including data from different ethnicities and regions would improve the generalizability of the findings. Future research should explore potential differences in dietary habits and aneurysm risk factors across different populations.
3. As suggested in the paper, gender and age can influence the occurrence of intracranial aneurysms (IAs). However, the current study did not perform such subgroup analyses. It would be beneficial to investigate the impact of dietary factors based on gender and age in future studies.
4. The dietary factors mentioned in the paper (e.g., fruit intake, lamb intake) are evaluated individually. However, analyzing overall dietary patterns and combinations of different foods may provide a more comprehensive understanding and lead to more robust conclusions.
5. The results of observational studies and the current MR analysis differ. Observational studies are often influenced by confounding factors, but comparing the results of different methods and discussing the differences in detail could deepen the understanding of these findings.
6. The paper suggests that diet influences intracranial aneurysms, particularly regarding fresh fruit intake. However, explaining the biological mechanisms through which fresh fruit consumption reduces IA risk would deepen the understanding of the results.
7. While MR analysis minimizes the impact of confounding factors, other potential confounders, such as lifestyle habits and genetic background, may still affect the results. Assessing the influence of these factors in future studies is important.
8. To more precisely evaluate the impact of diet, it would be helpful to use quantitative data on food intake, such as the amount and frequency of consumption. This would provide a clearer understanding of the relationship between dietary factors and intracranial aneurysms.
Author Response
Comments 1: The sample size of the GWAS databases used in the study is relatively small. Therefore, further research should use larger datasets to enhance the reliability of the results.
Response 1: Thank you for your comment. While we selected relatively large and recent datasets from the available GWAS data, we acknowledge that the sample size remains limited. To address this issue, larger GWAS datasets are needed, and we hope that future research will include such datasets to enhance the reliability and generalizability of the results. (Lines 233-237)
Comments 2: The study primarily focuses on data from European populations. Including data from different ethnicities and regions would improve the generalizability of the findings. Future research should explore potential differences in dietary habits and aneurysm risk factors across different populations.
Response 2: Thank you for your valuable suggestion. In response, we have included additional analysis of diet and aneurysms in Asian populations within the manuscript. This analysis further supports our conclusions, demonstrating that the protective effect of fresh fruit on aneurysms remains significant across different populations, and confirming the potential negative effects of lamb consumption. (Lines 162-166, Figure 4)
Comments 3: As suggested in the paper, gender and age can influence the occurrence of intracranial aneurysms (IAs). However, the current study did not perform such subgroup analyses. It would be beneficial to investigate the impact of dietary factors based on gender and age in future studies.
Response 3: Thank you for your insightful suggestion. During our analysis, we did consider the potential heterogeneity of results based on age and gender. Unfortunately, due to the lack of individual-level data in the GWAS datasets, we were unable to conduct subgroup analyses. We will add a corresponding note regarding this limitation in the manuscript. (Lines 237-240)
Comments 4: The dietary factors mentioned in the paper (e.g., fruit intake, lamb intake) are evaluated individually. However, analyzing overall dietary patterns and combinations of different foods may provide a more comprehensive understanding and lead to more robust conclusions.
Response 4: We greatly appreciate your suggestion. We agree that analyzing overall dietary patterns and combinations of different foods would provide a more comprehensive understanding. However, due to the limitations in available exposure data, we were unable to perform such an analysis in this study.
Comments 5: The results of observational studies and the current MR analysis differ. Observational studies are often influenced by confounding factors, but comparing the results of different methods and discussing the differences in detail could deepen the understanding of these findings.
Response 5: Thank you for your thoughtful comment. We have discussed the differences between our MR analysis and previous observational studies in the Discussion section. For instance, our study did not find an association between alcohol intake and aneurysm occurrence, which contrasts with the findings of some observational studies. (Lines 203-216)
Comments 6: The paper suggests that diet influences intracranial aneurysms, particularly regarding fresh fruit intake. However, explaining the biological mechanisms through which fresh fruit consumption reduces IA risk would deepen the understanding of the results.
Response 6: Thank you for your suggestion. While studies linking fresh fruit consumption and aneurysm risk are limited, we believe that antioxidant activity could be a potential biological mechanism. Fresh fruits are rich in antioxidant vitamins with anti-inflammatory properties, which may help reduce the production of pro-inflammatory eicosanoids, potentially preventing the formation and rupture of intracranial aneurysms.
Comments 7: While MR analysis minimizes the impact of confounding factors, other potential confounders, such as lifestyle habits and genetic background, may still affect the results. Assessing the influence of these factors in future studies is important.
Response 7: Thank you for your comment. Indeed, while MR analysis minimizes the impact of confounding factors, lifestyle habits and genetic background may still influence the results. However, MR uses instrumental variables (IVs), specifically single nucleotide polymorphisms (SNPs) strongly associated with the exposure, to investigate causality, thereby reducing the impact of confounders and reverse causality.
Comments 8: To more precisely evaluate the impact of diet, it would be helpful to use quantitative data on food intake, such as the amount and frequency of consumption. This would provide a clearer understanding of the relationship between dietary factors and intracranial aneurysms.
Response 8: Thank you for your suggestion. We did attempt to include quantitative data on food intake, such as the amount and frequency of consumption, to enhance the analysis. Unfortunately, suitable exposure data was not available for this purpose. We have added this limitation to the manuscript. (Lines 225-230)
Round 2
Reviewer 1 Report
Comments and Suggestions for Authors
n/a
Comments on the Quality of English Languagegood
Reviewer 3 Report
Comments and Suggestions for Authors
Although I have no formal reasons to doubt the approaches used in the work, I would still like to recommend that the authors critically review the methodology used. My doubts are related to the fact that the Authors cannot distinguish, for example, the effect of lamb consumption on the development of aneurysm and the effect of aneurysm development on food preference (lamb consumption).